# Public Knowledge, Beliefs and Attitudes toward the COVID-19 Vaccine in Saudi Arabia: A Cross-Sectional Study

**DOI:** 10.3390/healthcare10050853

**Published:** 2022-05-05

**Authors:** EL Walid EL Hassan, Amal Khaleel Abu Alhommos, Dalal Aliadhy, Shaima Alsalman, Ohoud Alnafaa, Ahmed Mohamed

**Affiliations:** Department of Pharmacy Practice, College of Clinical Pharmacy, King Faisal University, Al Ahsa 43518, Saudi Arabia; eelhassan@kfu.edu.sa (E.W.E.H.); ayase.ita@gmail.com (D.A.); shima3li.5300@hotmail.com (S.A.); ohoud.nafea@gmail.com (O.A.); aammohammed@kfu.edu.sa (A.M.)

**Keywords:** attitude, beliefs, COVID-19 vaccine, knowledge, Saudi Arabia

## Abstract

Background: With the help of vaccines, the world has witnessed a substantial decrease and even the eradication of many infectious diseases. Many factors influenced the public’s acceptance and compliance with the COVID-19 vaccine. Methods: This is a cross-sectional study that was conducted in Saudi Arabia. The questionnaire link was distributed through social media platforms. The questionnaire tool assessed the participants’ general knowledge and the barriers to taking the COVID-19 vaccine. All people who are currently living in Saudi Arabia and are at least 16 years old were included in the study and formed the study population. Based on our inclusion criteria, a total of 2,198 individuals were enrolled in this study. Results: Participants who were willing to take the vaccine were 68%. After vaccination, 10% of the participants think they can stop wearing face masks and washing their hands. Two-thirds of the participants think that the vaccine is effective in preventing COVID-19 infection. A total of 44.0% of the participants were afraid of the vaccine. The most common reasons were fear of unknown side effects (53.9%) and believing that the vaccine was not tested enough (27%). More than half of the study participants had a preference for receiving the Pfizer vaccine (52%). Conclusion: The public’s acceptance of the COVID-19 vaccine in Saudi Arabia is insufficient. Unknown side effects, widespread misconceptions, and a lack of adequate safety trials are all important reasons for concern among Saudis. More educational materials and awareness efforts will help to alleviate the fear that surrounds it. This will boost the vaccine’s acceptance rate in the long run.

## 1. Introduction

Throughout history, the world has witnessed a substantial decrease and even the eradication of many infectious diseases with the help of vaccines [1]. Smallpox is one of the most lethal diseases ever known, with a 30% mortality rate for the most common major form [2]. Three out of ten infected patients died because of the disease. In 1980, with the help of vaccination, the World Health Organization (WHO) announced the complete eradication of the disease. This was considered the most celebrated achievement in global public health [3]. The same case applies to polio, tetanus, hepatitis B, rubella, pertussis, and mumps, which the world has almost forgot even existed thanks to vaccination [4].

Coronavirus disease of 2019 (COVID-19) is an acute respiratory illness caused by SARS-CoV-2 that results in symptoms ranging from mild loss of taste or smell to severe complications or even death. The ongoing outbreak of COVID-19 was first observed in December 2019 in Wuhan, China [5]. The worldwide impact of COVID-19 has been estimated to cause a shrinkage of global growth by around 8% and cost the global economy around two trillion dollars in 2020. Because healthcare systems around the world were not prepared for this pandemic, they struggled to provide sufficient healthcare and some of them even collapsed [6].

According to a national report in the United Kingdom, in 2020, more people passed away due to COVID-19 than any other infectious disease since 1918 [7]. Since the first recorded case in Saudi Arabia in March 2020, a total of 388 thousand cases have been reported, with 6650 deaths [8]. Currently, there is no specific medication that could cure COVID-19. However, vaccines are offering a new approach to the prevention of COVID-19. Many pharmaceutical companies have got Emergency Use Authorization (EUA) approval for different vaccines with different mechanisms of action. The first COVID-19 vaccine was approved by the Food and Drug Administration (FDA) in December 2020 [9]. Multiple vaccines have been proven through multicenter randomized controlled trials to be safe and effective in decreasing the incidence and severity of COVID-19 infection [10].

Vaccine refusal has been a growing concern since the 1990s, especially after Andrew Wakefield linked autism to the measles-mumps-rubella (MMR) vaccine in one of his papers back in 1998. Since that publication, the health system has been facing many challenges regarding convincing the public about the safety and effectiveness of vaccines. The arguments and misconceptions regarding vaccination have changed over time. Prior to this, the major concerns regarding vaccines were the dangerous chemical ingredients like aluminum and antifreeze [11]. Currently, history repeats itself with the COVID-19 vaccine. A notable hesitancy was reported in many countries, and there has been a marked spread of anti-vaccination movements recently. This has caused concern among researchers all over the world [12,13,14].

Currently, many brands of the COVID-19 vaccine are marketed in different countries and are free of charge in some. However, the financial aspect is not the only factor that determines the public’s tendency to uptake the vaccine. The success of any vaccine depends in a large part on the public’s compliance with it [15]. Many factors collectively influence the public’s acceptance and compliance with the COVID-19 vaccine. Some of the main factors include the health status of the person and their family, religious factors, financial factors, the level of education, the field of employment, and conspiracies that are spread through social media platforms [16,17,18]. The most common misconception that circulates in the Saudi community is that the vaccine causes other conditions like facial nerve palsy, infertility, and genetic mutations. Some even think that receiving the COVID-19 vaccine can weaken immunity and cause infection. Despite the Saudi Ministry of Health’s denial of all of these myths, other fears prevent some Saudi Arabian residents from receiving the vaccine. Examples include fear of needles, fear of allergic reactions, unexpected side effects, teratogenic effects, and breastfeeding concerns. Some people reported uncertainty regarding the efficacy of the vaccine since it has been rushed, and some believe that it has not been tested enough [19,20]. Also, a conspiracy theory has been raised that the vaccine contains nanochips to locate and follow or even control people by governments.

Although research has reported some side effects, only 45 reports per million doses administered were considered serious, and 372 reports per million doses administered were non-serious adverse events. On the other hand, it has been reported that vaccination significantly reduced the rate of mortality caused by COVID-19 infection by 69.3%, with a 63.5% and 65.6% reduction in non-ICU and ICU hospitalization, respectively [21].

Public acceptance of the vaccine is influenced by many factors, with knowledge being one of the most important factors. In a previous study that assessed the Saudi general public’s knowledge regarding the seasonal influenza vaccination, it was estimated that a larger percentage of the surveyed participants had good knowledge and understanding of the flu vaccination [22]. In a published systematic review, the acceptance rate for the H1N1 influenza was found to range between 8% and 67% [23]. According to this systematic review, the acceptance rate in the United States was 64% [24], 59.5% in Hong Kong and China [25,26], and 56.1% in the United Kingdom [27]. In another study that aimed to assess COVID-19 vaccine acceptance before the release of the first vaccine, 64.7% of the surveyed participants were keen on getting the vaccine once it was available [28]. In this study, we included more regions of Saudi Arabia, with a total of 2000 participants. We added questions to assess reasons for fear or unacceptance of the COVID-19 vaccine. Whether the public trusted the government in terms of the vaccine was also assessed. The questionnaire was distributed after the release and provision of the vaccine in Saudi Arabia. While previous studies in Saudi Arabia focused on the clinical and psychological consequences of COVID-19 [29,30,31,32,33,34,35], few studies have investigated vaccination knowledge, attitudes, and practices. Besides, few studies have looked into COVID19 vaccine acceptability and its associated factors [36,37]. A study of health care workers (HCWs) in China discovered that the COVID-19 vaccination is very well accepted [37]. According to another study conducted in the United States, just 20% of people plan to refuse the COVID-19 vaccine, while 80% say they would take it [36]. Another global survey study that looked at COVID-19 vaccine acceptance rates across 19 countries found that 71.5% of participants reported that they would like to take the vaccine, with acceptance rates ranging from around 90.0% (in China) to 55.0% (in Russia) [38]. As vaccine acceptance is context-dependent and varies by culture and socioeconomic status, our study aimed to assess knowledge, beliefs, and attitudes toward the COVID-19 vaccine among the Saudi Arabian population. More specifically, the study answered the following research questions: What is the Saudi population’s knowledge, beliefs and attitudes toward the COVID-19 vaccine?Is there a difference in COVID-19 knowledge, beliefs and attitudes toward the COVID-19 vaccine based on gender, education, and occupation?

## 2. Materials and Methods

### 2.1. Study Design

A cross-sectional study was conducted between March and June 2021 utilizing an online questionnaire, which was distributed via social media channels. Different social media platforms were used to distribute the questionnaire, including Facebook, WhatsApp, Twitter, and Instagram. A convenience sampling technique was used to recruit the study sample. On the first page of the questionnaire, an informed consent form was provided, and participants had the choice to proceed or leave at that point. The study objectives were clearly mentioned to inform the participants about the importance of their participation. The use of online evaluation platforms is now considered acceptable research practice during pandemics, since a recent study indicated that their use increased considerably during the COVID-19 outbreak, notably for educational and research purposes. To reduce the possibility of selection bias, we shared the questionnaire link on various social media websites and pages with a diverse audience that was not limited to a specific demographic group (such as females, youth, or university students) in order to broaden the generalizability of our findings and include a diverse range of demographic groups.

### 2.2. Study Sample

The study population consisted of all people who are currently living in Saudi Arabia and are at least 16 years old. According to the most recent available statistics [39], Saudi Arabia’s total population is 35,844,909, with 24,374,538 people aged 15 and over accounting for 68.0% of the population. The minimum sample size required based on this calculation is 385 participants. We estimated that a sample size of 385 participants was required to provide 80% power of analysis with a 95% confidence interval using the formula n = N x/((N − 1)E2 + x).

Inclusion criteria:-All Saudi Arabians of all regions and nationalities.-Aged ≥ 16 years.

Participants who did not match the aforementioned criteria were excluded from the study.

Characteristics of the study participants: The number of participants was 2198. More than half of the participants (53.7%) were between the ages of 16 and 30 years old, and 68.2% of them were females. A total of 58.9% of them were from the Eastern area, and most of them were Saudis. More than half of the study participants (64%) were Bachelor’s graduates. The majority of participants (87.9%) were non-smokers. The nature of the jobs varied, with 32.6% being students. A total of 44.1% of the healthcare workers were pharmacists, (Table 1).

### 2.3. Instrument for Data Collection

The questionnaire, which was constructed based on a literature review, has 26 questions separated into five main sections [28,40,41,42,43] and common misconceptions among the Saudi Arabian public. Questions were reviewed and validated by professors from the College of Clinical Pharmacy at King Faisal University. They were questioned about the questionnaire’s clarity and comprehensibility and to explore its face validity, as well as whether any of the questions were difficult to comprehend. They were also asked if any of the questions were undesirable or offensive to them. They confirmed that the questionnaire was simple to comprehend and complete. Furthermore, a pilot study was conducted on a small group of participants to test the questionnaire’s comprehension before implementing it on a larger scale, and they affirmed that it is understandable and simple.

The first section of the questionnaire included basic demographic and occupation-related questions. The second section collected information regarding the medical history of participants. The third section aims to assess knowledge regarding the new vaccine, what it’s made of, how many shots it consists of, and who’s eligible to get it. The fourth section aimed to assess beliefs regarding the COVID-19 vaccine. In this section, we focused on common misbeliefs and concerns circulating in the community regarding the vaccine. Also, questions regarding myths and conspiracy theories were included in that section. The fifth and last section assessed the participants’ attitudes toward the vaccine, including their willingness to get vaccinated and their acceptance of the vaccine.

### 2.4. Statistical Analysis

SPSS (version 25.0 software (Chicago, IL, USA)) was used to analyse the data for this study. Categorical data were reported as percentages (frequencies). Chi-squared test/Fisher test was used as appropriate to compare proportions for categorical variables. Statistical significance was defined as a two-sided *p* < 0.05.

## 3. Results

The majority of the study participants (78.6%) didn’t suffer from any diseases, while 4.8% suffered from diabetes, 4.6% suffered from obesity, 4% suffered from multiple diseases, and 2.6% suffered from respiratory disorders. When the participants were asked whether they needed to get the vaccine after they recovered from COVID-19, 53.3% replied with “no”. Only 19% believed that they could stop wearing the face mask and washing their hands after getting the vaccine. The participants were asked a set of questions to reveal the extent of their knowledge about the COVID-19 vaccine, and their responses to these questions are presented in Table 2. The majority of the participants (84.9%) had good knowledge about the number of doses that should be taken for most COVID-19 vaccine brands. A total of 73.3% of the participants believed that the COVID-19 vaccine is useful for preventing infection with COVID-19. A similar percentage (74%) of the participants had good knowledge regarding the age groups that should receive the COVID-19 vaccine. A total of 83.3% of the participants have installed the Sehhaty application, and 83.5% knew how to register for the vaccine.

Table 3 demonstrates the participants’ knowledge of who shouldn’t receive the vaccine. A total of 73.2% of the participants believed that pregnant women should not take the vaccine. There were 43% who believed that the vaccine should not be given to breastfeeding mothers. A total of 52.6% believed that the vaccine should not be given to people who have a history of allergic reactions to any kind of vaccine. The amount of people who believed that the vaccine should not be given to the elderly over 65 years of age was 16.4%. There were 36.7% who believed that the vaccine should not be given to immunocompromised, HIV-positive, and cancer patients. A total of 51.5% believed that the vaccine should not be given to people under 16 years of age. Nearly one-quarter of the study participants (23.7%) believed that the vaccine should not be given to people with chronic medical conditions. A total of 16.3% believed that the vaccine should not be given to women of childbearing age.

Less than half of the study participants (41.7%) strongly agreed that everyone should take the vaccine. Around one-third (35.7%) the study participants didn’t agree to giving the vaccine to older adults with chronic diseases such as diabetes, hypertension, and respiratory diseases). A total of 43.8% of the participants had one or more fears related to the vaccine (Table 4). The two most important reasons for concern were found to be the fear of unknown side effects (53.1%) and the fear that the vaccine was rushed and not sufficiently tested (26.4%). There were 12.6% who reported being afraid of receiving the COVID-19 vaccine due to their fear of needles. Also, 11.6% did not trust pharmaceutical companies. A total of 7.5% believed that the vaccine caused the infection with COVID-19. Also, 6.5% believed that the vaccine could modify their genes, and 5.2% believed that the vaccine caused infertility in women. Less than half of the study participants (46.7%) believed that if they have already been infected with the virus, they do not need the vaccine. There were 81% of the participants who said they would not stop wearing a face mask and washing their hands even after receiving the vaccine.

More than half of them (68.1%) wanted the vaccine to protect themselves from infection, 56.7% wanted it to protect their family, 72.5% wanted it to protect society from infection, 14.1% wanted to be cured of the COVID-19 virus, and 17.9% believed it is vital for everyone to get vaccinated in order for the government to track the epidemic’s spread and manage it, Table 5.

Table 6 illustrates the level of acceptance of the COVID-19 vaccine among participants. We found that 68.5% desired to obtain the vaccine and that 69.3% would recommend the COVID-19 vaccine to their family and friends. In terms of vaccination types, 52.1% selected Pfizer BioNTech (American/German), 29.5 % preferred Johnson & Johnson (American), 16.6% preferred Oxford/AstraZeneca (British/Swedish), 1.1% preferred Sputnik V (Russian), and 0.7% preferred Sinopharm.

### Participants’ Demographics and Knoweldge about the Vaccine

There was no statistically significant difference based on gender for knowledge of the number of doses to be taken from the vaccine (*p* > 0.05), belief that the vaccine helps prevent COVID-19 virus (*p* > 0.05), and knowledge of how to register for the vaccine. While there were statistically significant differences based on gender for knowledge of the age groups that should take the vaccine (*p* < 0.05), and the installation of the Sehhaty app (*p* < 0.05), Table 7.

The results showed that there was statistically significant difference based on education and knowledge of the number of doses to be taken for the vaccine (*p* < 0.05), the belief that the vaccine is useful in preventing COVID-19 (*p* < 0.05), knowledge of the age groups that must take the vaccine (*p* < 0.05), the installation of the Sehhaty application (*p* < 0.05), and knowing how to register to get the vaccine, Table 8.

There were statistically significant differences based on the profession and knowledge of the number of doses to be taken from the vaccine (*p* < 0.05), the belief that the vaccine helps prevent COVID-19 infection (*p* < 0.05), knowledge of the age groups that must take the vaccine (*p* < 0.05), and the installation of the Sehhaty application and knowing how to register to get the vaccine (*p* < 0.05), Table 9.

## 4. Discussion

As far as we know, this current study is the first to measure the knowledge, beliefs and attitudes toward the COVID-19 vaccine after its release throughout all regions of Saudi Arabia. Furthermore, with a larger sample size of 2198, this study provides greater insight with further details on the situation in Saudi Arabia. Some key differences in this study are the investigation of reasons why the vaccine could be unlikable by some of the population, in addition to questions about knowledge regarding the COVID-19 vaccine and some misconceptions. Furthermore, a question was asked to see which vaccine the public tends to favor. 

Due to its high prevalence rate and life-threatening implications, the COVID-19 pandemic imposed significant emotional and social stress on several sectors of the community [44,45,46,47,48]. This enhanced the need for a variety of preventative interventions, including improving vaccination rates to boost community immunity and reduce prevalence. The rate of acceptance of the COVID-19 vaccine in our study is similar to that in a previous study in Saudi Arabia, which measured the determinants of acceptance of the vaccine and was conducted before the release of any COVID-19 vaccine. Willingness to take the vaccine was shown to be (68.5%), and similarly, it was (64.7%) in the compared study [28]. Another study in Oman reported a lower acceptance rate of (57.0%) [49]. Detoc et al. found that around 77% of the population would consent to receiving the vaccination [50]. When it comes to comparing attitudes across countries, studies have revealed significant differences. Gallè et al. conducted a previous study in Italy to assess the acceptance of COVID-19 vaccination among the elderly population and found a high acceptance rate (92.7%) [51]. This acceptance rate was higher than that reported by Del Riccio et al. [52] in another Italian study of a younger adult sample (81.9%). Another study looked at the influence of age on vaccine acceptability, finding that people above the age of 50 had higher acceptance than those under the age of 50 [53]. The main concerns expressed by the older participants were safety and efficacy [51]. It’s worth noting that one of the key elements influencing COVID-19 immunization adoption was the green pass. Domestic vaccine passports, according to a recent study, may have a negative impact on people’s motivation and readiness to be vaccinated [54]. To facilitate the safe resumption of social and economic activity following the COVID-19 epidemic, a mandatory green pass was proposed [51]. Simultaneously, it has been shown that required measures such as the adoption of a vaccine passport might be viewed as a threat to human rights and civil liberties, resulting in a drop in vaccine acceptance [55]. 

The desire to take the COVID-19 vaccine as soon as it became accessible in North America, South America, and Europe was demonstrated by Abdul et al. Panama received the largest percentage of positive replies (87.4%), while Russia received the lowest percentage (51.3%) [56]. Furthermore, the degree of robust compliance to COVID-19 vaccination as soon as it arrived in Africa, Asia, and Australia revealed that Australia had the largest proportion of positive responses (92.9%), while Egypt had the lowest proportion of responses (43.6%) in a recent systematic review of COVID-19 vaccine acceptance in different countries [57]. The acceptance rate reported in our study in Saudi Arabia is similar to that in other countries such as Russia (54.9%), Poland (56.3%), the US (56.9%), and France (58.9%). It was reported that the middle east and several other countries had the lowest rates of COVID-19 acceptance; for example, Kuwait’s acceptance rate was around 23.6%, and Jordan’s was around 28.4%. However, the acceptance rates of Saudi Arabia are not considered amongst the highest global acceptance rates, such as Ecuador (97.0%), Malaysia (94.3%), Indonesia (93.3%), and China (91.3%) (20). Low or not high enough acceptance rates are considered a major limitation to global healthcare efforts to control the pandemic of COVID-19. 

In a recent study that measured acceptance and hesitancy among low- and middle-income countries, the perceived effectiveness of vaccines among the public in the United States was 85%, whereas, in this study, it was 73.3% [58]. The most important reason to be afraid or reluctant to receive the vaccine is the fear of unknown side effects. In this study, 53.1% expressed fear of unknown side effects, which is much less than in the United States, which expressed 79.3% [58]. This result might be partly due to the exaggerated media coverage regarding the risk and incidence of adverse events such as thrombosis. Another reason for this excessive fear of unknown side effects goes back to the fact that the vaccine has been rapidly developed and released, as reported by 26.4% of participants. Some 9.9% were skeptical about the effectiveness of the COVID-19 vaccine, whereas, in the United States, the percentage of skeptical people was 46.8%. 

This confirms the findings of a previous study in Russia and the USA which reported a low level of confidence in the COVID-19 vaccine’s safety and efficacy [59]. Reported sources of mistrust among the participants in Russia and the USA were that the vaccine’s approval would be accelerated for political reasons (39.8%), they’d like to see additional safety and effectiveness data among them (32.7%), and they believe the vaccine is unsafe and could cause harmful side effects (28.4%) [59].

However, very few participants reported conspiracy theories (3.6%) or not trusting the pharmaceutical industry or government (11.6%), despite the exceptional attention dedicated by the media to such fears. When people were asked what brand of COVID-19 vaccine they would prefer to get, the most common answers were Pfizer/BioNTech (52.1%) and Oxford/AstraZeneca (16.6%), respectively. The logical reason why these two were the most commonly chosen brands is that they were the first two brands to arrive in the kingdom. The most probable reason why people preferred Pfizer over AstraZeneca is the adverse effect (thrombosis) that surrounded the AstraZeneca vaccine very early in its release. In addition, Pfizer’s effectiveness was perceived to be superior to that of AstraZeneca’s.

In our study, females showed better attitudes, practices, and knowledge compared to males in different aspects. This was confirming the findings of previous studies in Oman and Saudi Arabia which reported that females were shown to be more educated about nonpharmaceutical preventative measures and to have more positive behaviors and attitudes toward them [49,60]. There were statistically significant differences based on education and knowledge of the number of doses to be taken from the vaccine, belief that the vaccine is useful in preventing COVID-19, and knowledge of the age groups that must take the vaccine, according to our findings. Previous research has linked low literacy levels to a higher refusal rate for the COVID-19 vaccine [61,62,63]. Improving public awareness of the condition is critical for changing people’s attitudes and behaviors about it and lowering its prevalence [64].

The outcomes of this study will aid decision-makers in strategic planning and identifying vaccination target groups. In addition, reviewing vaccination knowledge will aid in identifying information gaps in the community and among certain groups, combating misinformation, and developing appropriate strategic management plans to improve the quality of vaccine acceptance.

This study has limitations. The generalizability of our findings may have been influenced by convenience sampling. Some demographic groupings may have been missing when data was collected via an online survey. This is evident given that nearly 90% of the participants in our study are under the age of 50 years. According to recent social media usage statistics, the vast majority of social media sites, such as Instagram, Facebook, Twitter, and Snapchat, are used by people under the age of 50 [65]. However, this was a recommended practice to reduce in-person contact and avoid disease transmission. Finally, without any further validation processes, we evaluated our newly developed (non-standardized) questionnaire instrument on a small group of people from the general population. Therefore, our findings should be interpreted carefully.

## 5. Conclusions

COVID-19 vaccination knowledge was found to be moderate among Saudis. Positive vaccine beliefs and a moderate level of vaccination acceptance were also demonstrated. Female gender, higher education level, and occupation type influenced participants’ knowledge, beliefs, and attitudes concerning the COVID-19 vaccine. The public’s acceptance of the COVID-19 vaccine in Saudi Arabia is insufficient. There were a number of common misunderstandings in society, which led to an unjustified fear of the vaccine. To reduce people’s fears and promote vaccine uptake, more educational materials and incentives to acquire the vaccine are required. 

## Figures and Tables

**Table 1 healthcare-10-00853-t001:** Participants’ demographic characteristics.

Variable	Frequency	Percentage
Age		
16–30 years	1180	53.7
31–50 years	783	35.6
51–65 years	214	9.7
>65 years	22	1.0
Place of residence in Saudi Arabia		
Eastern region	1295	58.9
Western region	214	9.7
Northern region	213	9.7
Southern region	213	9.7
Central region	264	12.0
Nationality		
Saudi	2051	93.3
Gender		
Female	1499	68.2
Male	700	31.8
Education		
Below high-school education	58	2.6
High school diploma	523	23.8
Bachelor’s degree	1408	64.0
Postgraduate	210	9.5
Smoker		
Yes	266	12.1
Occupation		
Health sector	261	11.9
Non-health sector	714	32.5
Not working	507	23.1
Student	717	32.6

**Table 2 healthcare-10-00853-t002:** The participants’ extent of knowledge of, and attitude towards the vaccine.

	Frequency	Percentage
How many shots of most brands of COVID-19 vaccine should be taken?		
Two shots	1867	84.9
One shot/three shots/I don’t know	332	15.1
Do you think that the COVID-19 vaccine is helpful in preventing COVID-19 infection?		
Yes	1612	73.3
No	218	9.9
I don’t know	369	16.8
According to your knowledge, which age groups should receive the vaccine?		
Older than 16 years old	1628	74.0
All ages/less than 16 years old/I don’t know	571	26.0
Have you ever installed Sehhaty application?		
Yes	1832	83.3
Do you know how to register for the vaccine?		
Yes	1837	83.5

**Table 3 healthcare-10-00853-t003:** The participants’ knowledge about who should not take the COVID-19 vaccine.

Who Do You Think Should Not Take the Vaccine?	Yes	
	Frequency	Percentage
Pregnant women	1610	73.2%
Breastfeeding women	946	43.0%
Allergy to previous vaccine	1156	52.6%
Elderly more than 65 years of age	360	16.4%
Immune compromised (HIV, cancer patients, or taking immune suppressing drugs)	807	36.7%
People less than 16 years of age	1133	51.5%
People with chronic medical conditions	522	23.7%
Women of childbearing age	358	16.3%
I don’t know	304	13.8%

**Table 4 healthcare-10-00853-t004:** The participants’ beliefs and worry regarding the vaccine.

	Frequency	Percentage
I believe that everyone should take the vaccine:		
Strongly agree	917	41.7
Agree	570	25.9
Neutral	454	20.6
Disagree	175	8.0
Strongly disagree	83	3.8
I believe that only elderly with chronic diseases (diabetes mellitus, hypertension, respiratory disease) should receive the vaccine:		
Strongly agree	364	16.6
Agree	288	13.1
Neutral	373	17.0
Disagree	786	35.7
Strongly disagree	388	17.6
I am afraid of receiving the vaccine:		
Yes	964	43.8

**Table 5 healthcare-10-00853-t005:** The participants’ motives behind receiving the vaccine.

Why Do You Think That Anyone Should Get the Vaccine?	Yes	
	Frequency	Percentage
To protect himself from future COVID-19 infection	1497	68.1%
To protect his family from future COVID-19 infection	1246	56.7%
To protect the community from future COVID-19 infection	1595	72.5%
To be cured of current COVID-19 infection	311	14.1%
I don’t believe that there is a valid reason to receiving COVID-19 vaccines	188	8.5%
To help the government locate the spread of the epidemic and enable them to control it.	394	17.9%

**Table 6 healthcare-10-00853-t006:** Participants acceptance of COVID-19 vaccine.

	Frequency	Percentage
I would get the COVID-19 vaccine?		
Yes	1507	68.5
I do not know	383	17.4
No	309	14.1
I would advise family and friends to get the COVID-19 vaccine?		
Yes	1525	69.3
I do not know	424	19.3
No	250	11.4

**Table 7 healthcare-10-00853-t007:** The knowledge and gender.

	Gender		*p*-Value
	Female	Male	
How many shots of most brands of COVID-19 vaccine should be taken?			
Two shots	1272 (57.8%)	595 (27.1%)	0.930
One shot/three shots/I don’t know	227 (10.3%)	105 (4.8%)	
Do you think that the COVID-19 vaccine is helpful in preventing COVID-19 infection?			
Yes	1096 (49.8%)	516 (23.5%)	0.271
No	141 (6.4%)	77 (3.5%)	
I don’t know	262 (11.9%)	107 (4.9%)	
According to your knowledge, which age groups should receive the vaccine?			
Older than 16 years old	1137 (51.7%)	491 (22.3%)	0.004
All ages/less than 16 years old/I don’t know	362 (16.5%)	209 (9.5%)	
Have you ever installed Sehhaty application?			
Yes	1223 (55.6%)	609 (27.7%)	0.002
No	276 (12.6%)	91 (4.1%)	
Do you know how to register for the vaccine?			
Yes	1245 (56.6%)	592 (26.9%)	0.372
No	254 (11.6%)	108 (4.9%)	

**Table 8 healthcare-10-00853-t008:** The relation between knowledge and gender.

	Education					*p*-Value
	No Education	Below High-School Education	High School Diploma	Bachelor’s Degree	Postgraduate	
How many shots of most brands of COVID-19 vaccine should be taken?						
2 shots	5 (0.2%)	24 (1.1%)	402 (18.3%)	1246 (56.7%)	190 (8.6%)	0.000
1 shot/3 shots/I don’t know	7 (0.3%)	22 (1.0%)	121 (5.5%)	162 (7.4%)	20 (0.9%)	
Do you think that the COVID-19 vaccine is helpful in preventing COVID-19 infection?						
Yes	5 (0.2%)	25 (1.1%)	372 (16.9%)	1047 (47.6%)	163 (7.4%)	0.000
No	0 (0.0%)	3 (0.1%)	59 (2.7%)	139 (6.3%)	17 (0.8%)	
I don’t know	7 (0.3%)	18 (0.8%)	92 (4.2%)	222 (10.1%)	30 (1.4%)	
According to your knowledge, which age groups should receive the vaccine?						
Older than 16 years old	3 (0.1%)	26 (1.2%)	363 (16.5%)	1073 (48.8%)	163 (7.4%)	0.000
All ages/less than 16 years old/I don’t know	9 (0.4%)	20 (0.9%)	160 (7.3%)	335 (15.2%)	47 (2.1%)	
Have you ever installed the Sehhaty application?						
Yes	6 (0.3%)	28 (1.3%)	399 (18.1%)	1219 (55.4%)	180 (8.2%)	0.000
No	6 (0.3%)	18 (0.8%)	124 (5.6%)	189 (8.6%)	30 (1.4%)	
Do you know how to register for the vaccine?						
Yes	6 (0.3%)	33 (1.5%)	409 (18.6%)	1213 (55.2%)	176 (8.0%)	0.000
No	6 (0.3%)	13 (0.6%)	114 (5.2%)	195 (8.9%)	34 (1.5%)	

**Table 9 healthcare-10-00853-t009:** Knowledge and profession.

	Occupation				*p*-Value
	Health Sector	Non-Health	Not Working	Student	
How many shots of most brands of COVID-19 vaccine should be taken?					
2 shots	248 (11.3%)	632 (28.7%)	388 (17.6%)	599 (27.2%)	0.000
1 shot/3 shots/I don’t know	13 (0.6%)	82 (3.7%)	119 (5.4%)	118 (5.4%)	
Do you think that the COVID-19 vaccine is helpful in preventing COVID-19 infection?					
Yes	220 (10.0%)	527 (24.0%)	328 (14.9%)	537 (24.4%)	0.000
No	21 (1.0%)	66 (3.0%)	56 (2.5%)	75 (3.4%)	
I don’t know	20 (0.9%)	121 (5.5%)	123 (5.6%)	105 (4.8%)	
According to your knowledge, which age groups should receive the vaccine?					
Older than 16 years old	210 (9.5%)	516 (23.5%)	362 (16.5%)	540 (24.6%)	0.027
All ages/less than 16 years old/I don’t know	51 (2.3%)	198 (9.0%)	145 (6.6%)	177 (8.0%)	
Have you ever installed the Sehhaty application?					
Yes	230 (10.5%)	648 (29.5%)	374 (17.0%)	580 (26.4%)	0.000
No	31 (1.4%)	66 (3.0%)	133 (6.0%)	137 (6.2%)	
Do you know how to register for the vaccine?					
Yes	234 (10.6%)	639 (29.1%)	393 (17.9%)	571 (26.0%)	0.000
No	27 (1.2%)	75 (3.4%)	114 (5.2%)	146 (6.6%)	

## Data Availability

The data presented in this study are available on request from the corresponding author.

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
