# Peer review of "Public Knowledge, Beliefs and Attitudes toward the COVID-19 Vaccine in Saudi Arabia: A Cross-Sectional Study"

_healthcare, 2022, doi:10.3390/healthcare10050853_

Round 1

Reviewer 1 Report

I am grateful for the opportunity to review this paper. The pandemic of COVID-19 has become a major threat to global public health. Therefore, identifying the level of public knowledge, and acceptance of the COVID-19 vaccine, is crucial to develop effective prevention and communication strategies resolving global economic crisis and social panic. In this context, the paper under review is aimed at assessing the Saudi Arabian population regarding perception, knowledge, acceptance, and fear of the COVID-19 vaccine.

The subject under study is important, especially in the historical period we are experiencing and the article presents interesting results but, it must be improved especially for its local impact and different epidemiological context in which it was carried out. I would like to encourage authors to consider several issues to be improved, in order to avid that the article will be yet another of those article that promises much but deliver little and therefore do not get cited.

Introduction: The authors should make clearer what is the gap in the literature that is filled with this study. The authors must better frame their study within the major literature evidences addressing the issue of COVID-19 vaccine acceptance in different groups of population. What is the international situation?

Methods: The survey was conducted using a non-standard questionnaire. The use of an unreliable instrument is a serious and irreversible limitation of the study. A validation process is mentioned but the results are not reported. The fact that the questionnaire has been revised by some experts is only an obvious preliminary requirement to create the tool. Nor the fact that a similar questionnaire has been used in previous surveys is sufficient. What about face validity, intelligibility, reliability? The enrolment procedure must be better specified. How was the sample enrolled? How did the authors choose the way to enroll the sample? How did they avoid the selection bias, considering the sampling methods? What is the reference population? And how large is it?

Statistical analysis: I suggest to insert a measure of the magnitude of the effect for the comparisons. Please consider to include effect sizes.

Discussion: I also suggest expanding, emphasizing what is the possible international contribution of the study compared with other evidences of literature. The discussion must be updated including comparison with similar studies carried on groups of population in the same period (refer to articles with DOI: https://doi.org/10.3390/vaccines9111222) and including the debated argument of the effects of a green pass linked to vaccination practice: a paragraph should be added with a proper reference.

Ethical issue: the Authors state that “The research ethics committee at the deanship of scientific 313 research at King Faisal University approved the study protocol". A valid approval number must be reported.

Author Response

Response to reviewers:

Reviewer 1:

I am grateful for the opportunity to review this paper. The pandemic of COVID-19 has become a major threat to global public health. Therefore, identifying the level of public knowledge, and acceptance of the COVID-19 vaccine, is crucial to develop effective prevention and communication strategies resolving global economic crisis and social panic. In this context, the paper under review is aimed at assessing the Saudi Arabian population regarding perception, knowledge, acceptance, and fear of the COVID-19 vaccine.

The subject under study is important, especially in the historical period we are experiencing and the article presents interesting results but, it must be improved especially for its local impact and different epidemiological context in which it was carried out. I would like to encourage authors to consider several issues to be improved, in order to avoid that the article will be yet another of those article that promises much but deliver little and therefore do not get cited.

- Introduction: The authors should make clearer what is the gap in the literature that is filled with this study. The authors must better frame their study within the major literature evidences addressing the issue of COVID-19 vaccine acceptance in different groups of population. What is the international situation?

- We would like to thank the reviewer for this valuable comment. We have now addressed it in page 4, lines 153-156 and lines 161-174.

Methods: The survey was conducted using a non-standard questionnaire. The use of an unreliable instrument is a serious and irreversible limitation of the study. A validation process is mentioned but the results are not reported. The fact that the questionnaire has been revised by some experts is only an obvious preliminary requirement to create the tool. Nor the fact that a similar questionnaire has been used in previous surveys is sufficient. What about face validity, intelligibility, reliability? The enrolment procedure must be better specified. How was the sample enrolled? How did the authors choose the way to enroll the sample? How did they avoid the selection bias, considering the sampling methods? What is the reference population? And how large is it?

- We would like to thank the reviewer for this valuable comment. As we mentioned in the method section under the sub-heading “Questionnaire development” we constructed our questionnaire tool based on previous literature to adapt it to the current pandemic situation and the culture of Saudi Arabia. We totally agree with the reviewer that having validated tool is a preferred option, however, as our questionnaire was mainly descriptive and asking the participants about their awareness, attitudes, and practices using MCQ format, not using 5-point Likert scale questions format, which in that case would need much more than revision by expert in the field.

Based on the reviewer comment, we have now provided further information about the efforts made to test the face validity of our tool and the small piloting that we conducted before distributing the questionnaire on a larger scale in the method section, page 7, lines 215-223. As we were not using 5-point Likert scale format questions we did not test the reliability of our scale for example using Cronbach's Alpha measure. In addition, we have mentioned the following limitations at the end of the discussion in page 17, lines 418-425 (The generalizability of our findings may have been influenced by convenience sampling. Some demographic groupings may have been missing when data was collected via an online survey. However, this was a recommended practice to reduce in-person contact and avoid disease transmission. Finally, without any further validation processes, we evaluated our newly developed (non-standardized) questionnaire instrument on a small group of people from the general population. Therefore, our findings should be interpreted carefully).

To reduce the possibility of selection bias, we shared the questionnaire link on various social media websites and pages with a diverse audience that was not limited to a specific demographic group (such as females, youth, or university students) in order to broaden the generalizability of our findings and include a diverse range of demographic groups, this is now highlighted in page 5, lines 184-190.

The study population consisted of all people who are currently living in Saudi Arabia and are at least 16 years old. This is now highlighted under the sub-heading “study sample”, lines 192-194.

According to the most recent available statistics [39], Saudi Arabia's total population is 35,844,909, with 24,374,538 people aged 15 and over accounting for 68.0 % of the population. The minimum sample size required based on this calculation is 385 participants. We have now added this information under the sub-heading “study sample”, lines 192-196.

Statistical analysis: I suggest to insert a measure of the magnitude of the effect for the comparisons. Please consider to include effect sizes.

- Thank you for this valuable comment. We have now addressed this comment.

Discussion: I also suggest expanding, emphasizing what is the possible international contribution of the study compared with other evidences of literature. The discussion must be updated including comparison with similar studies carried on groups of population in the same period (refer to articles with DOI: https://doi.org/10.3390/vaccines9111222) and including the debated argument of the effects of a green pass linked to vaccination practice: a paragraph should be added with a proper reference.

- We would like to thank the reviewer for this valuable comment. We have now addressed it in page 15, lines 358-367.

Ethical issue: the Authors state that “The research ethics committee at the deanship of scientific 313 research at King Faisal University approved the study protocol". A valid approval number must be reported.

- Thank you for this comment. We have now added the full ethical approval number.

Reviewer 2 Report

In this paper authors made a questionnaire tool in order to assess the participants’ general knowledge and the barriers to taking the COVID-19 vaccine, based in a  cross-sectional study that were conducted in Saudi Arabia. This  questionnaire link was distributed through social media platforms: WhatsApp, Twitter, and Instagram.

The results obtained show that the 44.0% of the particiants were afraid of the vaccine.

In this sense the most common reasons were fear of unknown side effects (53.9%) and believing that the vaccine was not tested enough (27%). More than half of the study 20 participants had a preference for receiving the Pfizer vaccine (52%).

The authors conclue that COVID-19 vaccine  acceptance among the Saudi Arabian public is not high enough.

The study is well planned and supported by a solid statistical analysis. However, this reviewer finds some limitations that can give a false picture of the results.

Basically the limitation could be found in the use of social media applications especially aimed at young population groups. This is a circumstance clearly observed in the United States and in Europe.

In fact when we analyze the results of the article we observe in the demographic characteristics of the participants only 1% were over 65 years old and if we take into account the participants over 50 years old we would only reach 10% in total (Table 1).

This circumstance should be commented on in the article and analyzed in the discussion.

Do the authors consider the possibility of using other types of tools and not just social networks? Do you consider the possibility of using other means such as Facebook, more used among older people?

Author Response

Response to reviewers:

Reviewer 2:

In this paper authors made a questionnaire tool in order to assess the participants’ general knowledge and the barriers to taking the COVID-19 vaccine, based in a cross-sectional study that were conducted in Saudi Arabia. This questionnaire link was distributed through social media platforms: WhatsApp, Twitter, and Instagram.

The results obtained show that the 44.0% of the participants were afraid of the vaccine.

In this sense the most common reasons were fear of unknown side effects (53.9%) and believing that the vaccine was not tested enough (27%). More than half of the study 20 participants had a preference for receiving the Pfizer vaccine (52%). The authors conclude that COVID-19 vaccine acceptance among the Saudi Arabian public is not high enough.

The study is well planned and supported by a solid statistical analysis. However, this reviewer finds some limitations that can give a false picture of the results. Basically the limitation could be found in the use of social media applications especially aimed at young population groups. This is a circumstance clearly observed in the United States and in Europe.

- We would like to thank the reviewer for this valuable comment. The use of online evaluation platforms is now considered acceptable research practice during pandemics, since a recent study indicated that their use increased considerably during the COVID-19 outbreak, notably for educational and research purposes. To reduce the possibility of selection bias, we shared the questionnaire link on various social media websites and pages with a diverse audience that was not limited to a specific demographic group (such as females, youth, or university students) in order to broaden the generalizability of our findings and include a diverse range of demographic groups. We have now highlighted this point in the method sections, page 5, lines 183-190 and at the same highlighted it in the limitations section in page 17, lines 418-425.

In fact when we analyze the results of the article we observe in the demographic characteristics of the participants only 1% were over 65 years old and if we take into account the participants over 50 years old we would only reach 10% in total (Table 1). This circumstance should be commented on in the article and analyzed in the discussion.

- We would like thank the reviewer for this comment. We have now highlighted and discussed this point in the discussion section in the limitation paragraph, page 17, lines 418-425.

Do the authors consider the possibility of using other types of tools and not just social networks? Do you consider the possibility of using other means such as Facebook, more used among older people?

- Thank you for this comment. The use of online evaluation platforms is now considered acceptable research practice during pandemics, since a recent study indicated that their use increased considerably during the COVID-19 outbreak, notably for educational and research purposes. To reduce the possibility of selection bias, we shared the questionnaire link on various social media websites and pages with a diverse audience that was not limited to a specific demographic group (such as females, youth, or university students) in order to broaden the generalizability of our findings and include a diverse range of demographic groups. However, this was a recommended practice to reduce in-person contact and avoid disease transmission. We have now highlighted these points in the method section, page 5, lines 184-190 and the limitations section, page 17, lines 418-425.

We are sorry we did not mention the Facebook among the used platforms despite that we have used it (we have now added it to the manuscript).

Reviewer 3 Report

Dear Authors,

Here attached, please find my comments!

Author Response

Response to reviewers:

Reviewer 3:

 The manuscript aimed to report on a cross-sectional study of the Saudi Arabian population regarding their awareness, attitudes, and practices of the COVID-19 vaccine. The manuscript title consists of a new topic that helps to clarify the state of public awareness, attitudes, and practices of the COVID-19 vaccine in Saud Arabia. However, the manuscript has several issues that do not comply with the international standards of writing a manuscript for international readers.

- This manuscript did not have research questions. Also, there is conceptual fuzziness of basic terms such as awareness and attitude and a lack of methodological rigor in the scale development, validation, and data analysis. When the substantive contents of the manuscript reviewed are tested against a standard scale construction and validation process, it did not fulfill the standards. For example, who drafted and validated the items was stated very generally.

- We would like to thank the reviewer for these valuable comments. We have now add the research questions for this study which reflect our aim under the sub-heading (2.4. Research questions) in page 6. Besides, we have now clearly highlighted that we developed the questionnaire tool based on previous literature and clarified that we developed our own questionnaire tool to adapt it to the current pandemic situation and the culture of Saudi Arabia. We totally agree with the reviewer that having validated tool is a preferred option, however, as our questionnaire was mainly descriptive and asking the participants about their awareness, attitudes, and practices using MCQ format, not using 5-point Likert scale questions format, which in that case would need much more than revision by expert in the field.

Based on the reviewer comment, we have now provided further information about the efforts made to test the face validity of our tool and the small piloting that we conducted before distributing the questionnaire on a larger scale in the method section, page 7, lines 215-223. As we were not using 5-point Likert scale format questions we did not test the reliability of our scale for example using Cronbach's Alpha measure. In addition, we have mentioned the following limitations at the end of the discussion in page 17, lines 418-425 (The generalizability of our findings may have been influenced by convenience sampling. Some demographic groupings may have been missing when data was collected via an online survey. However, this was a recommended practice to reduce in-person contact and avoid disease transmission. Finally, without any further validation processes, we evaluated our newly developed (non-standardized) questionnaire instrument on a small group of people from the general population. Therefore, our findings should be interpreted carefully).

- Page 3, in the methods section, lines 124-125, says, 'Questions were reviewed and validated by professors from the College of Clinical Pharmacy at King Faisal University.' This was insufficient to provide information about the validity evidence. The fact that a professor has examined was not enough compared with the procedural tasks to be accomplished in the item development and validation.

- Thank you for this valuable comment. Based on the reviewer comment we have provided further information about the efforts made to test the face validity of our tool and the small piloting that we conducted before distributing the questionnaire on a larger scale in the method section, page 7, lines 215-223.

- Scale development and validation are critical in the health, social, and behavioral sciences. The scale construction includes pre-testing the questions, administering the survey, reducing the number of items, and understanding how many factors the scale captures. This is usually followed by another phase, scale evaluation, to test the number of dimensions, reliability, and validity. I encourage the authors to read Boateng, Neilands, Frongillo, Melgar-Quiñonez, and Young (2018) for more details on scale development and validation. The absence of these may be part of the reasons, contributing to the authors' lack of rigor in data analysis.

- Thank you for this valuable comment. We totally agree with the reviewer regarding the above mentioned point. However, as we were not using 5-point Likert scale format questions we did not test the reliability of our scale for example using Cronbach's Alpha measure. In addition, we have mentioned the following limitations at the end of the discussion in page 17, lines 418-425. Our questionnaire tool was mainly descriptive using MCQ format to explore Saudi population's current perception, knowledge, acceptance, and fear of the COVID-19 vaccine. Besides, as we mentioned above we have highlighted further in the method section that we tested the face validity of our tool and did a small pilot study before distributing the questionnaire on a larger scale in the method section, page 7, lines 215-223.

.

- Also, the keywords presented in the title do not align with the concepts presented in the methods section. For example, on page 3, line 121, the 'questionnaire development section' consists of different components than those found on the title page. The authors stated that the questionnaire has five sections. However, they did not indicate which of these sections belong to the awareness, attitudes, and practices. Similarly, the third component says 'knowledge and awareness.' This description is not aligned with the title, which says 'awareness.'

- Thank you for this valuable comment.  Based on the reviewer's comment, we've modified the title and keywords to reflect the purpose of our study, which was investigated utilizing our questionnaire tool.

- We mentioned the five sections and what is measured under each section as follows in the method section on page 7, lines 224-231 “The first section of the questionnaire included basic demographic and occupation-related questions. The second section collected information regarding the medical history of participants. The third section aims to assess knowledge regarding the new vaccine, what it's made of, how many shots it's made of, and who's eligible to get it. The fourth section aimed to assess beliefs regarding the COVID-19 vaccine. In this section, we focused on common misbeliefs and concerns circulating in the community regarding the vaccine. Also, questions regarding myths and conspiracy theories were included in that section. The fifth and last section assessed the participants' attitudes toward the vaccine, including their willingness to get vaccinated and their acceptance of the vaccine”.

- I would encourage the authors to frame their conceptualization of ‘awareness, attitudes, and practices based on the literature. This framing is important to conceptualize the terms from a broader perspective for global significance. Also, the authors should use those conceptualizations consistently throughout the manuscript.

- We have now addressed the reviewer comment and used consistent terminology across the manuscript.

On page 11, lines 306 to 309, the first two sentences in the conclusions section deal with ‘The public's acceptance of the COVID-19 vaccine…and...several widespread misconceptions in the community, leading to an unreasonable fear of the vaccine.’ How do these statements resonate with the awareness, attitude, and practices? How can I compare the validity of these sentences without having research questions?

- In response to the reviewer's suggestion, we've updated these statements to better correspond with our research questions and study aim.

- The authors should fix several additional issues, if they want to submit this to another journal:

  1. Re-arrange the title, fitting words, and connectors. The problem with the title is the word ‘practices’ did not match with the connector ‘towards.’ I mean, there is no way to have ‘practices towards.’ Maybe ‘of’ or another similar word seems appropriate.

- We have now changed the study title to match our research questions and study aim.

  1. Abstract, methods section, first describe the study participants before proceeding to the study instrument. I suggest to move the study participants information found in the results section to the methods section. Also, this applies later in the manuscript methods section.

- We have now addressed the reviewer comment and moved the study participants’ information found in the results section to the methods section in the abstract and later in the method section, page 6.

Again, abstract, line 22, the phrase ‘not high enough,’ was not clear. Replace this phrase by a much clearer phrase. Why are you concerned about what it is not? I mean, you conducted the study to investigate what it is and to conclude based on that. Just describe what it is in a simple term.

- We have now rephrased the sentence to reflect that the acceptance level was insufficient as the following “The public's acceptance of the COVID-19 vaccine in Saudi Arabia is insufficient.”

The last sentence in the abstract, lines 23-26, I did not understand what the authors wanted to say. What was that? It cannot be a conclusion. Again, it cannot be a result either. Please, include a key concluding remark or maybe a study implication. In my view, the one you stated there does not give sense.

- We have now rephrased this paragraph to address the reviewer comment as the following “Unknown side effects, widespread misconceptions, and a lack of adequate safety trials are all important reasons of concern among Saudis. More educational materials and awareness efforts will help to alleviate the fear that surrounds it. This will boost the vaccine's acceptance rate in the long run”.

  1. The research questions are the questions around which you centre your research. I did not see the research questions. This was the main problem with this manuscript.

- We would like to thank the reviewer for these valuable comments. We have now add the research questions for this study which reflect our aim under the sub-heading (2.4. Research questions) in page 7.

  1. There is a conceptual inconsistency throughout the manuscript. For example, there was no clear operational definitions of concepts such as awareness and attitude.

- We have now addressed the reviewer comment and used consistent terminologies across the manuscript to reflect our aim.

  1. Page 3, lines 101 to 104, re-write the study aim, which says ‘to provide essential information for the assessment of the Saudi Arabian population… does not give sense to the international reader. Consider revising the aim with the international reader in mind.

- We have addressed the reviewer comment and rephrased the study aim as the following “our study aimed to assess knowledge, beliefs and attitudes toward the COVID-19 vaccine among the Saudi Arabian population”.

  1. Your study design was clear from the title of the manuscript and the contents in the abstract, but you did not include that in the methods section of the manuscript. Hence, on page 3, line 107, include a statement that indicates the cross-sectional nature of the study design.

- We have now addressed the reviewer comment and rephrased the sentence to be “A cross-sectional study was conducted between March and June 2021 utilizing an online questionnaire, which was distributed via social media channels”, page 5.

  1. Page 3, line 118, the first exclusion criterion does not give sense. Improve that criterion in a more meaningful way.

- We have now addressed the reviewer comment and changed the sentence as the following “Participants who did not match the aforementioned criteria were excluded from the study”.

  1. Page 3, line 121, the sub-title ‘questionnaire development’ should be replaced by ‘instrument for data collection.’

- We have now addressed this comment, page 7.

  1. The statistical analysis described on page 3, lines 136 to 139, has no substantive content from the study context. Re-write this section. The main problem for this maybe you did not have the research questions. Also, consider using factor analysis as part of your data analysis for more rigorous analysis.

- Thank you for this comment. The statistical analysis part described what are the statistical analysis test performed in our study.

  1. Page 3, lines 141 to 146, the first paragraph in the results section should move to the methods section.

- We have now addressed this comment and moved it to the methods section.

  1. The results section consists of a mere description of results. Also, the table captions and the substantive contents in the tables did not match for most of the tables.

- We have now rephrased the table captions to match their content.

I think, the manuscript benefits from an extensive language editing.

- We have now checked the language of the manuscript.

Round 2

Reviewer 1 Report

The paper was improved following my personal deep major revision. All comments were addressed. I consider the paper of interest expecially in this epidemiological context and I suggest acceptance for publication.

Author Response

-The paper was improved following my personal deep major revision. All comments were addressed. I consider the paper of interest expecially in this epidemiological context and I suggest acceptance for publication.

We would like to thank the reviewer for his/her valuable comments, which improved our manuscript significantly.

Reviewer 3 Report

Dear authors,

Thank you for taking the time to revise the manuscript according to the comments.

Please, consider the following additional comments.

  1. Put the research questions next to the final purpose statement in the introduction.
  2. Change the first sentence in the research questions piece as follows.

Use this first sentence. 'More specifically, the study answered the following research questions,' instead of 'The purpose of this research is to find answers to the following research questions:'

     3. On page 8, check the formating consistency for Table 8.

      4. Your conclusions should be aligned with the research questions. In its current form, the conclusions did not align with the research questions. Re-write the conclusions in line with the research questions. Also, do not forget to include the implications as well.

Author Response

Thank you for taking the time to revise the manuscript according to the comments.

Please, consider the following additional comments.

  1. Put the research questions next to the final purpose statement in the introduction.
  2. Change the first sentence in the research questions piece as follows.

Use this first sentence. 'More specifically, the study answered the following research questions,' instead of 'The purpose of this research is to find answers to the following research questions:'

- Thank you for your valuable comment. We have now addressed the reviewer comment in page 3.

  1. On page 8, check the formating consistency for Table 8.

- We have now checked the formatting consistency for Table 8.

  1. Your conclusions should be aligned with the research questions. In its current form, the conclusions did not align with the research questions. Re-write the conclusions in line with the research questions. Also, do not forget to include the implications as well.

- Thank you for your valuable comment. We have now addressed the reviewer comment in page 12.